# Association of Fitness and Fatness with Clustered Cardiovascular Disease Risk Factors in Nigerian Adolescents

**DOI:** 10.3390/ijerph17165861

**Published:** 2020-08-13

**Authors:** Danladi I. Musa, Abel L. Toriola, Daniel T. Goon, Sunday U. Jonathan

**Affiliations:** 1Department of Human Kinetics and Health Education, Kogi State University, Anyigba 272102, Nigeria; 2Department of Sport, Rehabilitation and Dental Sciences, Tshwane University of Technology, Pretoria 0001, South Africa; toriolaal@tut.ac.za; 3Faculty of Health Sciences, University of forte Hare, East London 5201, South Africa; dgoon@ufh.ac.za; 4Department of Human Kinetics and Health Education, Ibrahim BadamasiBabangida University, Lapai 911101, Nigeria; sundayj@ibbu.edu.ng

**Keywords:** gender difference, cardiovascular disease, adolescents, fitness, health promotion

## Abstract

*Purpose*: This study examinedthe independent and joint association of fitness and fatness with clustered cardiovascular disease risk (CVDrs) in 11–18 year-old Nigerian adolescents. *Methods*: A hundred and ninety seven adolescents (100 girls and 97 boys) were evaluated forfitness, fatness and CVDrs. Fitness was evaluated with the progressive aerobic cardiovascular endurance run test while fatness was assessed using body mass index. A clustered CVDrs was computed from the standardized residuals of total cholesterol, high density lipoprotein cholesterol, Low density lipoprotein cholesterol, triglycerides, plasma glucose, systolic blood pressure, and diastolic blood pressure. Regression models controlling for waist circumference assessed the association of fitness and fatness with CVDrs. *Results*: Prevalence of clustered CVD risk was 7.1% (girls = 3.0%; boys = 4.1%). Based on risk factor abnormalities, 52.8% of participants had one or more CVD risk factor abnormalities with more boys (27.4%) affected. Low fitness was associated with clustered CVDrs in both girls (R^2^ = 9.8%, *β* = −0.287, *p* = 0.05) and boys (R^2^ = 17%, *β* = −0.406, *p* < 0.0005). Fatness was not associated with the CVDrs in both sexes. After controlling for all the variables in the model, only fitness (R^2^ = 10.4%) and abdominal fat (R^2^ = 19.5%) were associated with CVDrs respectively. Unfit girls were 3.2 (95% CI = 1.31–7.91, *p* = 0.011) times likely to develop CVD risk abnormality compared to their fit counterparts. The likelihood of unfit boys developing CVD risk abnormality was 3.9 (95% CI = 1.15–10.08, *p* = 0.005) times compared to their fit peers. *Conclusions*: Fitness but not fatness was a better predictor of CVDrs in Nigerian boys and girls. The result of this study suggests that any public health strategies aimed at preventing or reversing the increasing trends of CVD risk in adolescents should emphasize promotion of aerobic fitness.

## 1. Introduction

Epidemiological studies consistently demonstrate that clustering of cardiovascular disease (CVD) risk factors called the metabolic syndrome (MS) increases the risk of CVD, type 2 diabetes (T2DM) and some forms of cancer [1,2]. When biological risk factors aggregate in the same individual, there is a multiplicative effect [3], which from a clinical perspective is instructive. Although the clinical endpoints for CVD rarely exist in pediatric populations, the development of fatty streaks begins in childhood [4]. One methodological shortcoming in the study of metabolic syndrome in youth is the lack of unanimous definition of the disorder in this population. Consequently, the American Diabetic Association (ADA) and the European Association for the Study of Diabetes (EASD) recommend the use of a continuous value of MS score for research in the pediatric populations [5]. This approach has been used by several investigators [6,7]. Recent evidence has shown that several CVD risk factors track from childhood to adulthood [8,9,10]. From a public health perspective, it is necessary to identify the youth who may be susceptible to clustered risk for the purpose of early prevention.

Two modifiable behavioral risk factors that are known to be associated with MS and CVD in adolescence are fatness and fitness [8]. Studies examining the association among cardiorespiratory fitness (CRF), fatness and CVD risk factors have produced mixed results [11,12,13,14]. In a study involving a cohort of 192 Scottish adolescents, Buchan et al. [11] reported an independent association among adiposity, CRF and cardiometabolic risk irrespective of adjustment for either of the independent variables. In another study involving 3202 American adolescents, Kwon and Co-workers [12] concluded that CRF independent of fatness had beneficial effects on lipid profiles in girls and on lipid profiles, insulin metabolism and inflammation levels in boys. In a study involving 12 to 13 year-old Welsh adolescents, Thomas and others [13] found fatness to be independently related to several CHD risk factors including blood pressure and some lipids. The results of Bouziotas and colleagues [14] suggested that neither fitness nor fatness was associated with CHD risk among 12 year-old Greek children. Given these conflicting results from the literature, further investigation is warranted.

The transition during adolescence is a critical period of life in terms of physiological and behavioral changes, for instance habits related to unhealthy behavior like smoking, binge alcohol abuse, excessive use of social media amongst others occur [15,16], which if not well managed may consequently lead to pediatric non-communicable disease of lifestyle. Since it is during this period important behaviors, including those lifestyles related to health of adults are established, it is pertinent to inculcate the life-long habits such as regular exercise and healthy feeding which will safeguard future health prospects.

Compared to children from developed countries [13,17,18], Nigerian adolescents exhibit relatively low aerobic fitness profile, though they display optimal weight status. Like their counterparts in developed and developing countries, Nigerian youth are also exposed to unhealthy lifestyles and associated health risks such as unhealthy feeding habits exacerbated by proliferation of fast food outlets, impact of social media and other sedentary behavior, particularly in big cities. Such lifestyle factors increase their susceptibility to low fitness and overweight. Furthermore, there is paucity of epidemiological data on possible associations among lifestyle parameters and CVD risk factors in Nigeria. Added to this, manyof the studies regarding CVD risk factors in Nigeria have focused mainly on CVD risk factor profiles [19,20,21]. Thus, the purpose of the present study was primarily two-fold: First, to examine the independent association of CRF and BMI with clustered CVD risk among Nigerian adolescents; second to determine the joint association of CRF and BMI with clustered CVD risk. A secondary purpose of the study was to characterize the CVD risk factor profile of Nigerian adolescents. It was hypothesized that adolescents with low CRF (fitness) and high BMI (a surrogate indicator of fatness) will demonstrate adverse CVD risk. Since risk factors of CVD are known to track from adolescence to adulthood [8], any information on their relationships is potentially relevant in planning effective preventive strategies.

## 2. Methods

### 2.1. Participants

A cross-sectional school-based study comprising of 197 apparently healthy secondary school children (100 girls and 97 boys), ranging in age from 11 to 18 years drawn from two secondary schools in Kogi East Senatorial District, Kogi State of Nigeria consented to participate in the project. The study was conducted between September and November, 2019 and participants were selected using probability sampling procedure. Specifically, the students were systematically selected using the class register in which every 4th student, starting from the second learner on class list was selected to participate in the study. Participants were eligible to participate in the study if they had no history of CVD or other reported health problems. School athletes and students who were not in an overnight 12-h fasting state were excluded from the study. The purpose and procedures of the test were fully explained to participants after duly obtaining permission from the various school heads. Written informed consent of parents/guardians and assent of participants were obtained before the study. All tests were conducted from 9 to 12 h in accordance with the principles of Helsinki Declaration after prior approval was received from the Ethics Review Board of Kogi State University, Nigeria.

### 2.2. Physical Characteristic Measurements

Physical characteristics of participants were measured using the protocol of the International Society for the Advancement of Kinanthropometry (ISAK) [22]. Specifically, barefoot body mass and stature were measured indoors, in each school with the aid of an electronic weighing scale (Seca digital floor scale, Sec-880; Seca, Birmingham, UK) and a wall-mounted stadiometer (Model Sec-206; Seca). Subsequently, participants’ body mass index (BMI) was computed and expressed as weight in kilogram divided by stature in meters squared (kg·m^−2^). Body mass index (BMI) was used to estimate body fatness. Both the triceps and medial calf skinfold thicknesses were measured on the right side of participants’ bodies with the aid of the Harpenden skinfold calipers (Creative Health Products, Ann Arbor, MI, USA). All measurements were taken thrice and the median of the three readings recorded as the criterion measure. The revised regression equations of Slaughter and co-workers, as cited [23] for black children, were used to estimate % fat from the sums of triceps and medial calf skinfolds:

Participants were categorized into two groups based on their BMI values according to the FITNESSGRAM revised data [24]. Based on age-specific comparisons, boys with BMI values between 13.7 and 26.6 kg·m^−2^ or below were categorized as healthy weight (HW). The corresponding data for girls were 13.5–25.0 kg·m^−2^. Participants with values above the upper levels were considered overweight (OW). Details of the participants’ classification into healthy weight (HW) and overweight (OW) based on age-specific comparisons have been described elsewhere [25].

Waist circumference (WC), which estimates abdominal fat [26], was measured with a retractable metal tape (Creative Health Products) at the level of umbilicus and midway between the lower rib margin and the iliac crest to the nearest 0.1cm. Two measurements were taken and the average score recorded. All anthropometric measures were conducted by a trained ISAK-certified Level 2 anthropometrist (the lead author).

### 2.3. Fitness Testing

Cardiorespiratory fitness was assessed using the progressive aerobic cardiovascular endurance run (PACER) protocol. The PACER is a multi-stage aerobic capacity test adapted from the 20 m shuttle run test (20-MST) that progresses in intensity. The PACER is a widely used, valid, and reliable test of aerobic fitness in children and adolescents [27]. The number of laps completed by each participant was used to estimate their CRF [28]. In order to assess the relationship between CRF and CVD risk factors, the total sample was divided into two groups on the basis of each participant’s performance in the test according to FITNESSGRAM sex and age revised health-related cut-points [24]. For boys and girls, the cut-points of 23 to 94 and 7 to 61 laps were used as healthy fitness levels or zones (HFZ), respectively. Those achieving HFZ were categorized as ‘fit’,whereas, their counterparts with values below the benchmark were classified as ‘unfit’.

### 2.4. Biochemical Measurement

Finger stick blood samples were obtained after twelve hour overnight fast from participants by a trained laboratory technologist and two trained nurses. Fasting glucose (GLU), total cholesterol TCL), high density lipoprotein cholesterol (HDL), low density lipoprotein cholesterol (LDL) and triglycerides (TG) were obtained using finger stick blood samples, which were analyzed with a CardioCheck Plus Analyzer (CCPA) (PTS Diagnostics, Indianapolis, IN, USA). The reliability and validity of CCPA have been reported, with reliability coefficients ranging from 0.86 to 0.91 for TCL, HDL, TG and LDL [29]. Participants were asked to sit down for at least 10 min after which they took their turns for the test. The middle finger was cleaned with an alcohol wipe, after which a gentle pressure was applied to the finger with lancet and the finger stuck at the center. Thereafter, gentle pressure was applied to the finger to produce a large drop of blood which was drawn into a capillary tube until the sample reached the fill line of the tube. The capillary tube was then positioned above the blood application window of the equipment and squeezed gently to dispense the entire blood sample on the strip without allowing the capillary tube to touch the test strip. Test results were then displayed on the analyzer within 90 s. The same measurement procedure was repeated for all participants.

### 2.5. Blood Pressure Measurement

An automated digital BP monitor (HEM-705 CP; Omron, Tokyo, Japan) was used to measure BP after participants were quietly seated for 10 min. The resting systolic blood pressure (SBP), diastolic blood pressure (DBP) and pulse rate were monitored on each participant’s right arm using appropriate cuff sizes, and this instrument has been shown to be accurate [30]. Specific details of the BP protocol have been described [25].

### 2.6. Clustered Cardiovascular Risk Score

A clustered CVD risk score (CVDrs) was computed from the following variables: plasma GLU, SBP, DBP, TCL, LDL, HDL, and TG (some of which are components of MS). Each of these variables was standardized by subtracting the mean value for each sex group from the individual’s value and then dividing the score obtained by the value of standard deviation [z = (value − mean)/SD]. The standardized HDL was multiplied by −1 because it is inversely related to the CVD risk. The z-scores of the individual risk factor were summed to create a clustered CVD risk score (continuous variable) for each participant with a lower score indicating a more favorable CVD risk profile. A CVD clustering was defined as the presence of 3 or more CVD risk factors in an individual.

### 2.7. Thresholds for CVD Risk Abnormalities

Participants’ CVD risk abnormalities were determined using the International Diabetic Federation (IDF) [31] standards on the concentrations of the following biochemical constituents indicated in parentheses: TCL (≥5.20 mmol); LDL (≥3.38); TG (≥1.7 mmol); HDL (≤1.04 mmol); GLU (≥5.6 mmol); as well as SBP (≥130 mmHg); and DBP (≥85 mmHg). Participants without risk factor abnormalities were classified as “no risk” and those with one or more risks are classified as “risk”.

### 2.8. Data Analysis

Data analyses were conducted with the aid of the Statistical Package for the Social Sciences (version 20, SPSS Inc., Chicago, IL, USA) at an alpha level of 0.05. Descriptive statistics (mean ± SD) of measured and derived variables were used to summarize the sample’s characteristics. The independent samples *t*-test was computed to test for differences in physical characteristics, performance, and CVD risk profile between girls and boys. Zero-order correlation coefficients were used to assess the relationship between fitness, fatness and participants’ CVD risk factors. Multiple regression analyses were conducted to determine the independent and combined association of fitness, fatnessand theCVDrs. All analyses were adjusted for WC as a confounding variable. The independent association of fitness and fatness with CVD risk profile was determined with thelogistic regression model. Separate analyses were performed for girls and boys. Odd ratios of being at risk of CVD were calculated between fitness and fatnesscategories. The model was adjusted for WC as confounding variable. Because of absenteeism and incomplete data, 197 (100 girls and 97 boys) participants out of 206 completed the measurements and their data used in the statistical analyses. This amounted to a compliance rate of 95.6%.

## 3. Results

### 3.1. Physical, Physiological and Biochemical Characteristics

Participants’ general characteristics stratified by sex are presented in Table 1. On the average, boys displayed significantly (*p* < 0.05) lower body mass, BMI, body fat percent and abdominal fat than girls. Boys also performed significantly (*p* < 0.0005) better than the girls in the fitness test. There were no significant (*p* > 0.05) differences between the sexes in all the CVD risk factors and CVDrs. All the CVD risk factor values were generally lower than the internationally proposed standards by the IDF [31], and girls generally displayed poorer risk profile than boys. Only a small proportion of boys and girl could meet the FITNESSGRAM criterion reference standard for CRF with girls demonstrating superior performances. Both sexes had healthy BMI, but correlation coefficients between fitness and fatness were generally weak regardless of sex.

### 3.2. Prevalence of CVD Risk Abnormalities

The prevalence of CVD risk abnormalities are presented in Table 2. Approximately 53% of the participants(girls = 25.4%; boys = 27.4%) displayed CVD risk abnormalities. 

The most prevalent CVD risk abnormalities were those pertaining to hypoalphalipoproteinemia (22.8%), hyperglycemia(20.8%) and diastolic hypertension (12.2%). However, these are not mutually exclusive. A greater proportion of boys tended to suffer the adverse effect of glucose abnormality than girls, while girls suffer more from DBP abnormality than boys. The proportion of boys and girls with HDL abnormality was similar.

Table 3 shows the data of participants with different numbers of CVD risks. As indicated, 47.2% had no risk abnormality. Out of the 52.8% with abnormalities, some had one, two, three or more risks. Clustering of CVD was found in 7.1% (girls = 3.0; boys = 4.1) of participants with majority of cases being from the male group. When participants with one or more risk factors were compared with those without risk factors; that is, CVD risk profile (Table 4), the latter group was significantly (*p* < 0.0005) older, heavier (*p* = 0.16) and displayed substantially greater WC (*p* = 0.0005) in both sexes. There were no significant (*p* < 0.05) differences in TCL, and LDL. Significant (*p* < 0.05) differences were noted in HDL, TG, GLU, SBP, DBP, CVDrs and the PACER, all in favor of the group without risk in both sexes. It is noteworthy that the levels of the CVD risk factors even among the groups with risk are within acceptable health standards [31].

### 3.3. Predictors of Cardiovascular Disease Risk Score

Multiple regression analyses were conducted to examine the independent association of fitness and fatness with CVDrs. Fitness was the independent predictor of CVDrsin both girls (*p* = 0.005) and boys (*p* < 0.0005) while fatness was not significantly (*p* > 0.05) associated with the dependent variable in either sex. In girls, fitness explained only 9.8% (F_2,97_ = 5.299, *p* = 0.007; *β* = −0.287, *p* = 0.005) of the variance in the dependent variable whereas in boys, fitness explained 17% (F_2,94_ = 9.647, *p* < 0.0005; *β* = −0.406, *p* < 0.0005) of the variance in the dependent variable.

Because, abdominal fat had greater significant (Girls: r = 0.551, *p* = 0.01; Boys: r = 0.382; *p* = 0.05) relationship with CVDrs than fitness and fatness in both girls and boys, the models for both sexes were adjusted for this variable. Results of the multiple regression models assessing the ability of the independent variables to predict CVDrs after controlling for abdominal adiposity are presented in Table 5. In girls, the covariate explained 19.5% of the variance in CVDrsin step 1. The addition of fitness and fatness in step 2 increased the total variance cumulatively to 20.4%, thus indicating the both independent variables explained an additional variance of only 0.9% (F_3,96_ = 8.20, *p* > 0.05) after controlling for the covariate. Abdominal fat was the only significant predictor (*p* < 0.0005). The model for boys explained 20.4% (F_3,93_ = 7.965, *p* < 0.0005) with 10.4% contribution from fitness and fatness. Fatness did not make any unique contribution. Fitness made a greater statistically significant (*β* = −0.279, *p* = 0.015) contribution than abdominal fat (*β* = 0.245, *p* = 0.049) (see Table 5 for details).

Results of the logistic regression models indicated that in general, only abdominal adiposity and fitness displayed significant effect, which was greater in boys than in girls. In girls, only fitness (OR = 3.2, 95% CI = 1.31–7.91; *p* = 0.011) was negatively associated with the CVD risk profile. Adding WC did not improve the girls’ model but did for boys. The boys’ model was also significant (*p* < 0.0005), and both fitness (OR = 3.9, 95% CI = 1.51–10.08; *p* = 0.005) and abdominal fat (OR = 9.1, 95% CI = 1.06–77.78; *p* = 0.044) were substantially associated with the dependent variable, respectively.

## 4. Discussion

Recentcross-sectional and longitudinal studies in Western societies have shown that overweight and unfit adolescents display adverse CVD risks compared to their healthy weight and fit peers [12,32]. Despite this evidence, the importance of fitness and fatness in the development of these disorders has not been fully explored among African youth, especially Nigerian adolescents.

Although, it is well known that PACER and BMI are related to CVD or MS risk factors in youth [7], a number of these studies suffered some methodological shortcomings mainly due to lack of a unified definition of childhood MS or CVD risk, as they used dichotomous classifications for each variable [5,7], but there is increasing evidence in support of the use of a clustered MS or CVD score which is more sensitive and less error prone [6,32]. Thus, the use of continuous CVDrs used in this study is appropriate.

The result of this study shows that clustering of CVD risk factors exist among participants, and it is more prevalent in boys than girls. Similar situation has been observed among the Northern Ireland youth [33]. The prevalence of CVD risk of 7.1% documented in this study is relatively higher than the 5.4% observed in Danish youth [34] and 5% Iranian youth [35]. But the prevalence rate of 11.9% observed in nine-year-old Norwegian children [6] is strikingly higher than ours. Potential reasons for this high prevalence rate, despite their younger age, may not be unconnected with consumption of energy-laden food which is characteristic of youth from Western societies−a probable consequence of high obesity rates [36]. For the individual CVD risk, hypoalphalipoproteinemia was most prevalent in girls and hyperglycemia, most rampant in boys.

This study indicated that fitness and fatness are weakly related to the dependent variable, although the relationship with fitness is stronger, particularly in boys. This result may be partly due to the low prevalence of overweight in both sexes and poor performance in CRF, especially among the boys in our sample. Previous research has shown that, in general, the correlation among fitness, fatness and CVD risk factors are low to moderate among samples of children and adolescents with normal weight [37]. Despite the weak associations, the link is still important in health terms.

In general, the mean values of the CVD risk factors in our sample are generally lower than the cut-point values established by the IDF [31]. This may not be surprising as majority of youth in our sample live in rural areas where the people are predominantly peasant farmers and artisans-occupations which are labor-intensive. Moreover, the staple food in the area includes grains, root and stem tubers with lots of vegetables and sparse fast foods.

Findings from this study showed that the mean scores for all risk factors of CVD (except for HDL) were in most cases significantly (*p* < 0.05) greater in participants with one or more CVD risks compared to those without any risk (Table 4). The pattern was similar in both sexes. This has also been observed in previous reports [6,32].

Findings from this study clearly indicate that fitness but not fatness was the independent predictor of CVDrs in boys and girls. Our results are in agreement with those of Buchan and co-workers [11] but at variance with some previous reports [13,18]. In a cross-sectional study involving 192, 14–16 year-old Scottish adolescents, Buchan et al. [11] observed a negative correlation between $\dot{V}$O_2_ and cardiometabolic risk after adjustment for confounding variables. Boreham and colleagues [18] reported that the observed relationship between fatness and CHD risk status was independent of fitness, and that fatness was a stronger predictor of CHD risk status than fitness among adolescents in Northern Ireland. Potential reasons for the inconsistencies in results may pertain to disparity in age, measurement protocols used and sampling variations.

Our results clearly indicated that the joint contribution of fitness and fatness in predicting CVDrs was moderate (9%), but only fitness made a unique contribution. The major determinants of CVDrs were abdominal fat and fitness. Abdominal fat uniquely explained20% of the variance in CVDrs among girls and 11.6% in boys after controlling for the covariate. Our results are in agreement with those of Buchan et al. [11] who found WC and CRF to be strongly associated with cardiometabolic risk in Scottish youth after controlling for sex, age and sexual maturity. These results have important health implications. For instance, it has been observed that the association between WC and health outcomes may be explained by its strong relationship with visceral abdominal fat, which is widely known to be a strong predictor of many diseases including dyslipidemia, insulin resistance, hypertension and all-cause mortality in youth and adults [38,39]. Our findings highlight the important need for targeting reduction in fatness, particularly abdominal adiposity and increasing CRF levels in adolescents, especially boys in order to minimize cardiometabolic risks in this population.

The present findings suggest that high levels of aerobic fitness are associated with better cardiometabolic health in adolescents with a clustering of CVD risk factors. Our results have also demonstrated that adverse effects of low aerobic fitness on CVD risk are more clearly observed in boys than girls. Based on our results and those of others, there is convincing evidence that fitness is an important risk factor that independently contributes to the development of CVD. Evidence from the present study and others should promote the development of fitness-driven public health strategy to reduce cardiovascular disease risk in adolescents.

However, findings from this study should be interpreted in the light of a number of limitations. First, the findings are constrained by the cross-sectional design used for data collection which precludes confirmation of causality among the variables. Secondly, non-inclusion of sexual maturity, a variable known to influence fitness test results in youth, also limits our understanding of the role of maturation in the relationship between the dependent and independent variables investigated in this study. However, a major strength of this study lies in its use of health-related cut-points for estimating aerobic fitness and fatness. This approach showed that adolescents who met the FITNESSGRAM CRF and BMI standards had better CVD risk profile than those who failed to attain such standards.

This study has shown that clustering of CVD risk factors exists among Nigerian adolescents. Fitness but not fatness is independently associated with CVDrs in participants. The relationship between fitness and CVDrs is stronger in boys than in girls. The joint contribution of fitness and fatness in predicting CVDrs is moderate, but abdominal adiposity and fitness were the major determinants of cardiovascular risk among Nigerian adolescents.

## Figures and Tables

**Table 1 ijerph-17-05861-t001:** Physical, biochemical and hemodynamic characteristics of participants according to sex (n = 197).

Variable	Girls (n = 100)	Boys (97)	*t*-Value	*p*
Age (y)	14.7 ± 2.2	14.6 ± 2.2	0.232	0.816
Stature (cm)	159.6 ± 7.3	160.3 ± 11.9	0.452	0.652
Body mass (Kg)	55.5 ±12.2	50.4 ± 12.5	2.744	0.007
BMI (kg·m^−2^)	21.6 ± 4.0	19.3 ± 2.6	4.679	0.0005
Body fat (%)	19.8 ± 7.4	11.2 ± 3.0	10.804	0.0005
TCL (mmol)	3.6 ± 0.8	3.4 ± 0.7	1.174	0.242
HDL (mmol)	1.3 ± 0.3	1.4 ± 0.5	1.235	0.219
LDL (mmol)	2.2 ± 0.6	2.1 ± 0.7	1.665	0.098
TG (mmol)	1.1 ± 1.2	0.9 ± 0.4	1.204	0.231
GLU (mmol)	5.0 ± 0.7	5.1 ± 0.7	1.183	0.238
WC (cm)	66.9 ± 9.4	64.3 ± 8.0	2.098	0.037
SBP (mmHg)	106.9 ± 17.0	103.5 ± 16.0	1.446	0.150
DBP (mmHg)	69.7 ± 15.5	69.8 ± 13.4	0.054	0.957
MAP (mmHg)	82.1 ± 15.0	80.9 ± 12.7	0.613	0.541
CVDrs	−7.3 ± 3.2	−8.1 ± 2.9	1.924	0.056
PACER (laps)	25.1 ± 13.4	38.8 ± 16.7	6.364	0.0005
Passed CRF	53 (53)	46 (47.4)		
Passed BMI	80 (80)	94 (96.9)		
r between CRF and BMI	−0.243	−0.099		

Numbers in parentheses are percentages.

**Table 2 ijerph-17-05861-t002:** Prevalence of CVD risk abnormalities by sex (n = 197).

Risk Factor	Criteria	Girls (n = 100) n %		Boys (n = 97) n%		Total (197) n%	
TCL (mmol)	≥5.20	1	1.0	1	1.0	2	1.0
HDL (mmol)	≤1.04	22	22.0	23	23.7	49	22.8
LDL (mmol)	≥3.38	4	4.0	4	4.1	8	4.1
TG (mmol)	≥1.70	9	9.0	4	4.1	13	6.6
GLU (mmol)	≥5.60	17	17.0	24	24.7	41	20.8
SBP (mmHg)	≥130	7	7.0	3	3.1	10	5.1
DBP (mmHg)	≥85	17	17.0	7	7.2	24	12.2

**Table 3 ijerph-17-05861-t003:** Participants with different number of CVD by sex (n = 197).

Group	0	1	2	3	Total
Girls	46 (46.0)	30 (30.0)	18 (18.0)	6 (6.0)	100 (50.8)
Boys	47 (48.5)	30 (30.9)	12 (12.4)	8 (8.2)	97 (49.2)
Total	93 (47.2)	60 (30.4)	30 (15.2)	14 (7.1)	197 (100.0)

**Table 4 ijerph-17-05861-t004:** Characteristics of participants according to presence of CVD risks (n = 197).

Characteristic	Girls (n = 100)			Boys (n = 97)		
	No Risk (n = 46)	Risk (n = 54)	*t*-Value	No Risk (n = 47)	Risk (n = 50)	*t*-Value
Age (y)	13.6 ± 2.2	15.6 ± 1.8	4.911 **	13.6 ± 1.9	15.6 ± 2.0	5.001 **
Body mass (kg)	52.5 ± 12.9	57.6 ±11.2	2.089 *	48.7 ± 12.0	52.0 ± 12.8	1.310
BMI (kg·m^−2^)	20.9 ± 4.2	22.1 ± 3.8	1.570	18.7 ± 2.1	19.9 ± 2.8	2.387 *
WC (cm)	62.9 ± 7.0	70.4 ± 9.9	4.427	59.5 ± 3.9	68.9 ± 8.3	7.218 **
TCL (mmol)	3.6 ±0.9	3.6 ± 0.8	0.111	3.5 ± 0.7	3.4 ± 0.8	1.075
HDL (mmol)	1.4 ± 0.2	1.2 ± 0.3	4.725 **	1.5 ± 0.5	1.2 ± 0.4	3.198 *
LDL (mmol)	2.3 ± 0.5	2.2 ± 0.7	0.393	2.0 ± 0.5	2.2 ± 0.8	1.320
TG (mmol)	0.7 ± 0.3	1.3 ± 1.6	2.691 *	0.8 ± 0.3	1.0 ± 0.4	2.272 *
GLU (mmol)	4.8 ± 0.6	5.2 ± 0.8	2.691 *	4.8 ± 0.4	5.5 ± 0.9	4.887 **
SBP (mmHg)	99.5 ± 15.3	113.3 ± 16.0	4.369	96.7 ± 15.0	109.9 ± 14.2	4.471 **
DBP (mmHg)	62.4 ± 12.0	75.7 ± 15.7	4.585 **	64.1 ± 10.5	74.8 ± 14.1	4.192 **
CVDrs	−9.0 ± 2.5	−5.8 ± 2.9	5.781	−9.6 ± 2.1	−6.7 ± 2.9	5.630 **
PACER (lap)	29.7 ± 13.3	21.0 ± 12.4	3.396 **	46.7 ± 14.2	31.2 ± 15.5	5.132 **

* *p* < 0.05 ** *p* < 0.01.

**Table 5 ijerph-17-05861-t005:** Relationship between fitness and fatness in a generalized linear model of CVDrs.

Group	Variable	Model 1		Model 2	
		*β*	*p*	*β*	*p*
Girls	WC	0.441	0.0005	0.433	0.0005
	BMI	-	-	−0.079	0.445
	PACER	-	-	−0.079	0.476
Boys	WC	0.347	0.002	0.245	0.049
	BMI	-	-	−0.203	0.051
	PACER	-	-	−0.279	0.015

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
