# Peer review of "Association of Fitness and Fatness with Clustered Cardiovascular Disease Risk Factors in Nigerian Adolescents"

_ijerph, 2020, doi:10.3390/ijerph17165861_

Round 1
Reviewer 1 Report
Thank you for the opportunity to review this paper. The authors investigate the association between CRF and BMI with cardiovascular risk factors in a group of adolescents in Nigeria.
- Abstract, although some of these comments are explained in other sections of the paper, they should be clarified in the abstract
- The inverse of HDL concentrations was used to calculate the risk score
- The definition of CVD is explained only for the continuous variable. However, the authors report the prevalence of CVD risk scores in the abstract and it is not clear what are the cutpoints
- The authors mention regression models in their abstract, but they don't report ß coefficients, I suggest they do
- Intro
- Fatty streaks are described as atherosclerotic sequelae. More like anatomical changes?
- Lines 64-65 “Lifestyle is described as a decision, I would hesitate to say so. Although it might be true that lifestyle is established in adolescence, I would not call it a decision, since there are multiple external factors that determine someone’s behavior”.
- Methods
-
- 2.3 Fitness testing: CRF is estimated in VO2? More details about the categorization should be explained
- 2.4 Can the authors provide some information about the reliability and validity of these tests?
- 2.6 This section is very clear. I wonder if the authors can show (table or figure preferably (histograms)) the distribution of the variables that contribute to the continuous cvd risk score they calculated. Additionally, it would be interesting to see the distribution of this variable (also histogram). Have the authors considered categorizing participants using the continuous variable? (Using the median as threshold?)
- 2.7 Suggest to use a different word than abnormalities, perhaps elevations? "Each of the factors included in the clustered cv risk score was dichotomized…"
- 2.8 The t-tests performed compare boys and girls, but the main comparison groups of the paper are fit vs unfit and normal-weight vs overweight. I suggest that authors use t-tests to compare participants characteristics by fitness and fatness status rather than by sex.
- The following phrase needs clarification "The Chi-square contingency analysis was used to examine the association between fitness, fatness and CVD risk profile.” It is not clear what authors are evaluating with the Chi-square.
- When authors mention that they evaluate the independent and combined associations, it is not clear how they did this. (Interactions? Fit and normal weight compared to fit an ow, unfit and normal weight, and unfit and overweight?)
- Authors talk about the CVD risk profile, but it is not clear if they refer to the continuous measure (sum of z-scores) or what they refer to as “abnormalities”. They mention that the association between fitness and fatness with CVD risk profile was determined with logistic regression, but is the CVD risk profile is a continuous measure, the logistic regression is not appropriate.
- Why was waist circumference used as a confounder in the analysis?
- Results
- Approximately 53% of the participants (girls = 25.4%; boys = 27.4%) this is not clear from Table 2.
The text in this paragraph 3.2. Prevalence of CVD risk abnormalities does not match table 2. (For example, in the text it says “hyperglycemia (20.8%), but the table shows 22.3%)
Table 4 shows “Risk-free” column, but this is not defined. Also, the notion of risk-free is not the best way to name this group (better to call, absence of risk factors or something in those lines)
An interesting finding that waist circumference was still associated with cardiovascular risk score after controlling for BMI. I suggest that authors highlight this mo
- Approximately 53% of the participants (girls = 25.4%; boys = 27.4%) this is not clear from Table 2.
Author Response
REVIEWER 1
Abstract
- The cut points are shown on pp. 4, lines 168-173:
Thresholds for CVD risk abnormalities
Participants’ CVD risk abnormalities were determined using the International Diabetic Federation (IDF)[30] standards on the concentrations of the following biochemical constituents indicated in parentheses: TCL (≥5.20 mmol); LDL (≥3.38); TG (≥1.7mmol); HDL (≤1.04mmol); GLU (≥5.6mmol); as well as SBP (≥130 mmHg); and DBP (≥ 85 mmHg. Participants without risk factor abnormalities were classified as “no risk” and those with one or more risks are classified as “risk”
- The coefficients have been indicated in the abstract. See line 24.
Low fitness was associated with clustered CVDrs in both girls (R2=9.8%,= -0.287, P =0.05) and boys (R2=17%, = -0.406, P<0.0005).
Introduction
- The statement has been rephrased, pp1, lines 41-42
Although the clinical endpoints for CVD rarely exist in pediatric populations, the development of fatty streaks begins in childhood [4].
- The word decision has been changed to behaviors (line 67) as suggested by the reviewer.
Methods
- Fitness testing- more detailed description have been provided. See pp.3, lines 134-136.
For boys and girls, the cut-points of 23 to 94 and 7 to 61 laps were used as healthy fitness levels or zones (HFZ), respectively. Those achieving HFZ were categorized as ‘fit’ whereas, their counterparts with values below the benchmark were classified as ‘unfit’.
- Reliability and validity of CardioCheckPlus Analyzer (CCPA). Some Chinese investigators have reported reliability coefficients ranging from 0.86-0.91 for TCL, HDL, TG and LDL. Reference provided:
Gao Y, Zhu CG, Wu NQ, Guo YL, Liu G, Dong Q, Li JJ. Study of reliability of CardioCheckPlus Analyzer for measuring lipid profile. Journal of Peking University (Health Sciences) 2016, 48(3):523-528.
See lines 142-144.Thank you for the suggestion.
The reliability and validity of CCPA have been reported with a reliability coefficients ranging from 0.86-0.91 for TCL, HDL, TG and LDL [29].
- The word elevation may apply to other lipids but not HDL. We still believe abnormality (or abnormal levels) seems appropriate.
- The chi square statistics has been expunged since it did not add anything new to the results.
- We used multiple regression models to determine independent association of the independent variables (IV) with the dependent variable (DV) and hierarchical regression model for the combined association of the IV with the DV.
- The CVD risk profile is the dichotomous variable used for the logistic regression. The two categories are: participants without risk (no risk) and those with one or more risks (risk). See pp.6, line 215.
When participants with one or more risk factors were compared with those without risk factors, that is, CVD risk profile (Table 4).
- WC was used as a cofounder because, of all the IV, it correlated most highly with CVDrs.
- Risk-free has been rephrased to read ‘no risk’ and those with risk as ‘risk’.
- Information on pp.6, lines 212-215 have been reconciled with that in Table 2. Thank you for this important observation.
Table 3 shows participants with different number of CVD risks. As indicated, 47.2% had no risk abnormality. Out of the 52.8% with abnormalities, some had one, two, three or more risks. Clustering of CVD was found in 7.1% (girls = 3.0; boys =4.1) of participants with majority of cases being from the male group.
Thank you.
Reviewer 2 Report
The manuscript was well thought out and presented a wealth of information, however I found the description of the statistical analysis and the reporting of results very difficult to understand. I am unsure if the authors over presented information (i.e. t-statistics and F-statistics that are typically not reported) or if there was a lack of understanding about the correct statistical method to use. Some specific examples:
- The authors indicate that the patient characteristics reported in table 1 were compared between boys and girls using either a student's t-test or a chi square test. Was normality of the continuous characteristics assessed before the use of a t-test? In my experience many lipid profile values are not normally distributed (often very skewed). If that is the case, a student's t-test is not appropriate. Instead a Wilcoxon rank sum test should be used. This is particularly relevant to triglyceride levels and PACERS laps where the reported SD is nearly have the magnitude of the mean. This is typically a clear indication that the distribution is not normally distributed.
- Page 7; line 234
I am very confused by the following statement. "In girls, only fitness (OR = 3.2, 95% CI = 1.31-7.91; p = 0.011) was
negatively associated with the risk of CVD, and the model explained 21.6 to 28.9% of the variance in CVDrs.". The authors state that logistic regression was used, assuming the dependent variable was a dichotomous outcome of risk or risk-free. However the second part of the statement refers the variance in CVDrs explained by the model and indicates a negative association. According to the authors, CVDrs is a continuous risk score. So how does a logistic model explain the variance in a continuous outcome? And how is an odds ratio on 3.2 interpreted as negative? - Page 7; line 236
The authors state the following: "The model for boys was also significant (p <0.0005), and both abdominal
fat (OR =9.1, 95% CI = 1.06-77.78;p =0.044) and fitness (OR = 3.9, 95% CI =1.51-10.08; p =0.005) were positively and negatively associated with the dependent variable respectively." Since both odds ratios are greater than 1.0, I don't understand why the OR for abdominal fat is referred to as positive and the OR for fitness refered to as negative? - The authors seem to have taken the component lipid parts, transformed them into a z-score and are calling that the CVDrs. This CVDrs was then used to define those who had risk and those who were risk-free. The description of how risk and risk-free were defined is not entirely clear and needs to be described with more detail.
The authors performed several multiple linear and logistic regression models but seem to be mixing the results from those models. Analysis of the CVDrs (a continuous outcomes) and analysis of CVD risk group (at risk vs. risk-free) use linear and logistic regression, but they are testing different hypotheses.
My suggestion would be to have a statistician assist the authors in presenting the results more clearly so that the reader than interpret them.
A few other minor comments:
- Throughout the manuscript, p-values are indicated with a greek symbol for rho. While rho does in fact look like a "p", I found it very comfusing with the population parameter defined by rho. I suggest a manuscript-wide edit to replace the Greek letter "rho" with the Roman letter "p".
- Page 3, line 104
The authors express BMI correctly in notation form, but when written out in the sentence describe it as "weight in kilograms squared divided by stature in meters". It should be "weight in kilograms divided by stature in meters squared". - Page 3; line 112
"health" should be "healthy" - Page 4' line 160
the words "were conducted" are repeated twice
There are several instances beyond those I specifically mention above where the syntax of the sentences need to be corrected.
Author Response
REVIEWER 2.
- Thank you for this important observation on test of normality especially for TG and PACER. Preliminary analysis actually indicated some degree of skewness but with our large sample size, we felt this will not affect the data adversely.
- Thanks for the observation. We have used the phrase CVD risk profile to name the dichotomous variable used to calculate logistic regression. The CVD risk profile is made up of participants with one or more risks and those without a single risk. This variable is therefore different from the clustered risk score which is continuous. See pp.6, line 215.
When participants with one or more risk factors were compared with those without risk factors, that is, CVD risk profile (Table 4),the latter group was significantly (P<0.0005) older, heavier (P =0.16) and displayed significantly greater and WC (P =0.0005) in both sexes.
- This query has been addressed on pp.7, lines 245-250. The presentation has been rephrased appropriately.
Results of the logistic regression models indicated that in general, only abdominal adiposity and fitness displayed significant effect, which was greater in boys than in girls. Models for both sexes were adjusted for WC. In girls, only fitness (OR = 3.2, 95% CI = 1.31-7.91; P =0.011) was negatively associated with the risk of CVD. The model for boys was also significant (P<0.0005), and both abdominal fat (OR =9.1, 95% CI = 1.06-77.78; P =0.044) and fitness (OR = 3.9, 95% CI =1.51-10.08; P =0.005) were associated with the dependent variable respectively.
- It is not CVDrs that was used to categorize participants into absence of risk and presence of risk but the CVD risk profile as explained above.
- The appropriate symbol has been used to replace the Greek letter both in the text and abstract. Thank you.
- The BMI formula has been corrected. See pp. 3, line 108. Thank you.
Subsequently, participants’ body mass index (BMI) was computed and expressed as weight in kilograms divided by stature in meters squared (kg.m-2).
Many thanks for the useful suggestions
Reviewer 3 Report
1. The authors did not define clustered CVD risk. Is that two or more CVD risk factors in one person a clustered?
2. The descriptions in the results part are confusing. This part should be reorganized. For example,
“Results of the logistic regression models indicated that in general, only abdominal adiposity and fitness displayed significant effect, which was greater in boys than in girls. Models for both sexes were adjusted for WC.” The WC reflects abdominal adiposity, right? Then why WC was adjusted when examined the association between abdominal adiposity and CVDrs?
What is the outcome (dependant variable in the model) in this sentence? I suppose is CVDrs or not. The authors should make the report clear.
3. Line 20-21: “Prevalence of clustered CVD risk was 7.1% (girls=25.4%; 21 boys=27.4%). “
I could not find this in your Result part. If the prevalence in girls is 25.4%, in boys is 27.4%, then the overall prevalence should between 25.4% and 27.4%. Why the number is 7.1%?
4. Line 199-200: “Clustering of CVD was found in 7.1% (girls = 3.0; boys =4.1) of participants with majority of cases being from the male group.”
Similar to the last comment, I cannot find and understand these numbers.
5. Line 24-25: “After controlling for WC, only fitness (R2=10.4%) and abdominal fat (R2=19.5%) were negatively and positively associated with CVDrs respectively. “
Please rephrase this sentence.
6. Line 236-238:
“The model for boys was also significant <0.0005), and both abdominal fat (OR =9.1, 95% CI = 1.06-77.78; =0.044) and fitness (OR = 3.9, 95% CI =1.51-10.08; =0.005) were positively and negatively associated with the dependent variable respectively.”
This description is not correct, please rephrase it.
Author Response
REVIEWER 3.
- The clustered CVD has been defined. See pp. 4, line167. Thank you.
A CVD clustering was defined as the presence of 3 or more CVD risk factors in an individual.
- Results of logistic regression have been clarified. See pp.7, lines 245-250. This has also been reflected in the Abstract.
Results of the logistic regression models indicated that in general, only abdominal adiposity and fitness displayed significant effect, which was greater in boys than in girls. Models for both sexes were adjusted for WC. In girls, only fitness (OR = 3.2, 95% CI = 1.31-7.91; P =0.011) was negatively associated with the risk of CVD. The model for boys was also significant (P<0.0005), and both abdominal fat (OR =9.1, 95% CI = 1.06-77.78; P =0.044) and fitness (OR = 3.9, 95% CI =1.51-10.08; P =0.005) were associated with the dependent variable respectively.
- The figures have been reconciled with those in the Table. See Table 2 on pp.6. Line 211
Table 2. Prevalence of CVD risk abnormalities by sex (n = 197).
Risk factor |
Criteria |
Girls (n = 100) n % |
Boys (n= 97) n % |
Total (197) n % |
|||
TCL (mmol) |
≥ 5.20 |
1 |
1.0 |
1 |
1.0 |
2 |
1.0 |
HDL (mmol) |
≤ 1.04 |
22 |
22.0 |
23 |
23.7 |
49 |
22.8 |
LDL (mmol) |
≥ 3.38 |
4 |
4.0 |
4 |
4.1 |
8 |
4.1 |
TG (mmol) |
≥ 1.70 |
9 |
9.0 |
4 |
4.1 |
13 |
6.6 |
GLU (mmol) |
≥ 5.60 |
17 |
17.0 |
24 |
24.7 |
41 |
20.8 |
SBP (mmHg) |
≥ 130 |
7 |
7.0 |
3 |
3.1 |
10 |
5.1 |
DBP (mmHg) |
≥ 85 |
17 |
17.0 |
7 |
7.2 |
24 |
12.2 |
- These figures have been reconciled both in the Abstract Section and in the result Section. PP 6, lines 212-215.
Table 3 shows participants with different numbers of CVD risks. As indicated, 47.2% had no risk abnormality. Out of the 52.8% with abnormalities, some had one, two, three or more. Clustering of CVD was found in 7.1% (girls = 3.0; boys =4.1) of participants with majority of cases being from the male group.
- The sentence has been rephrased and clarified. See Abstract
- As stated above, results of the logistic regression have been clearly presented both in the Abstract and in the Results Section.
- Many thanks for your useful observations.
Round 2
Reviewer 2 Report
Thank you for clarifying the categorization of risk and clearly defining the differences between CVDrs and CVD risk profile. The manuscript is now much easier to understand.
I believe that the correct statistical tests were used within the manuscript now that the outcomes have been better described. I don't necessarily agree that the "large sample size" makes the problem of applying parametric tests to skewed data go away; however as I do not have the data in hand, I will take the author's word that the skewness was minor enough to justify the use of a parametric test.
Author Response
- Thank you for your favorable comments.
- Thank you again for your understanding in endorsing our position.
Reviewer 3 Report
- Page 6, line 228-229, the authors said “while fatness was not significantly (P<0.05) associated with the dependent variable in both sexes”. But the P<0.05 here means it is significant. Please correct it.
- Page 7, line 245-247, “Results of the logistic regression models indicated that in general, only abdominal adiposity and fitness displayed significant effect, which was greater in boys than in girls. Models for both sexes were adjusted for WC.”
The authors did not answer my question raised last time:
The WC reflects abdominal adiposity, right? Then why WC was adjusted when examined the association between abdominal adiposity and CVDrs? Will this cause over-adjusted?
3. The language in the results part needs to be revised to make sure the grammars and organizations are right and concise.
Author Response
- The first query on non-significant (p>0.05) association of fatness with the dependent variable in both sexes has been addressed. The correct expression has been indicated (p.7, line 231). Thank you for this important observation.
- The second observation has also been addressed. We did this by recasting the whole paragraph to make the presentation clearer. See p.7, lines 248-253.
- Results of the logistic regression models indicated that in general, only abdominal adiposity and fitness displayed significant effect, which was greater in boys than in girls. In girls, only fitness (OR = 3.2, 95% CI = 1.31-7.91; P =0.011) was negatively associated with the CVD risk profile. Adding WC did not improve the girls’ model but did for boys. The model for boys was also significant (P<0.0005), and both fitness (OR = 3.9, 95% CI =1.51-10.08; P =0.005) and abdominal fat (OR =9.1, 95% CI = 1.06-77.78; P =0.044) were substantially associated with the dependent variable.
- For the issue on language, we have also tried our best to ensure conciseness.
- The authors wish to thank you immensely for your thoroughness and contribution to the improvement of the quality of our manuscript.